# The Effects of Core Machining Configurations on the Mechanical Properties of Cores and Sandwich Structures

**DOI:** 10.3390/ma15020521

**Published:** 2022-01-10

**Authors:** Zhiwen Qin, Lili Wei, Mingming Zhang, Rui Zhang, Xiang Ji, Xiaofei Song, Shengguan Qiang

**Affiliations:** 1National R&D Center of Wind Turbine Blade, Institute of Engineering Thermophysics, Chinese Academy of Sciences, Beijing 100190, China; mmzhang@iet.cn (M.Z.); 18340310600@163.com (X.J.); sxiaofei1126@163.com (X.S.); qiangshengguan@iet.cn (S.Q.); 2Key Laboratory of Wind Energy Utilization, Chinese Academy of Sciences, Beijing 100190, China; 3NMG Composites Co., Ltd., Jiaxing 314511, China; lili@nmgonline.com (L.W.); rain@nmgonline.com (R.Z.)

**Keywords:** core machining configurations, sandwich structures, grooves, perforated holes, contour cut, mechanical properties

## Abstract

Composite sandwich structures are widely used in the fields of aviation, marine, and energy due to their high specific stiffness and design flexibility. Improving the mechanical properties of the cores is significant to the strength, modulus, and stability of composite sandwich structures. Two kinds of core machining configurations were designed by combining thin grooves, perforated holes, and thick contour cuts as well as non-machining plain cores. The cores and sandwich structures with these configurations were fabricated using a vacuum-assistant infusion process. Static tensile, compressive, shear, and peeling tests were conducted on the infused cores and sandwich structures. The results showed that the tensile, compressive, and shear moduli, and compressive strength of the infused cores can be greatly improved. The tensile strength changed negligibly due to stress concentration induced by irregular foam cell and the shear-lag phenomenon of the resin column/foam interface. The shear strength of the infused cores increased slightly. The thick contour cuts and perforated holes can greatly improve the face sheet/core peel capacity of the sandwich structures, whereas the thin grooves can moderately improve the peel capacity. Both infused cores with the designed machining configurations exhibited positive effects on the compressive, tensile, and shear moduli, and compressive strength, considering the material costs. The study provides a comprehensive and quantitative insight into the effects of core machining configurations on mechanical properties of infused cores and composite sandwich structures.

## 1. Introduction

Composite sandwich structures that include two composite face sheets outside and a thick and light core inside are widely applied in various engineering fields due to their high stiffness and low weight. For a typical composite sandwich structure, the face sheets carry the most in-plane loads, and the core contributes a high bending stiffness by large core thickness. With the help of adhesive bonding between the face sheet and core material, the sandwich structure combining the face sheets and core functions as a whole to fulfill various requirements of engineering components.

However, the traditional composite sandwich structures still have weaknesses in terms of low strength of the face sheet/core interface and out-of-plane properties. A few techniques have been proposed and widely adopted to modify the architecture of the core alone or of sandwich structure, such as z-pinning, stitching, and core machining with holes and channels, to overcome these drawbacks.

Regarding the z-pinning technique, the materials with higher strength or modulus compared with core materials are selected for the pins to enhance the performance of composite sandwich structures. The pins are generally orthogonal or at an inclined angle to the face sheet of the sandwich structures. This technique provides an alternative to improve various mechanical properties of the composite sandwich, including the stiffness, strength, and other structural properties [1]. Marasco et al. [2] tested the out-of-plane tensile, compressive, and shear stiffness, and strength of K-Core/X-Core pin reinforced foam core sandwich panels. The results showed that the specific stiffness of sandwich panels was higher than that of the conventional Nomex sandwich cores, but the strength was lower. Du et al. [3] proposed a new z-pinned foam core with a truss network of pultruded carbon fiber pins for sandwich beams. The results indicated that the shear moduli and bending stiffness of sandwich beams increased with the increase in the volume fraction, the modulus, the diameter, and the inclination angle of the pins. Nanayakkara et al. [4] experimentally investigated the effect of orthogonal z-pins on the flat-wise compression properties and behavior of composite sandwich structures. The results revealed that the compressive modulus of the sandwich structure grew substantially with the volume content of z-pins. The compressive stress and total absorbed compressive strain energy of the sandwich structure increased by nearly 700% and 600%, respectively, compared with the traditional sandwich structures. Abdi et al. [5] experimentally studied the effect of the polyester pin reinforced polyurethane foam core on the flexural and flatwise compression behavior of the composite sandwich panel. The findings showed that the flatwise compressive load/weight and flexural stress and load/weight of the panel were significantly improved. Furthermore, these properties upgraded with the increase in the diameter of polymer pins. Zhou et al. [6] experimentally studied the carbon and glass composite pins for cross-linked PVC foam on the compressive and energy-absorbing characteristics of sandwich panel. The output revealed that both the compressive strength and the energy-absorbing capability of the reinforced foam panels were much higher than their plain foam counterparts. The author suggested that the combination of a thin and stiff pin and a medium-density foam is an optimal option that can offer outstanding compressive strength and high energy absorption.

Regarding the stitching technique for the reinforcement of composite sandwich structures, high strength or modulus fibers are commonly used to penetrate the core and the face sheets. Apparently, this approach can strengthen the bond between the face sheets and the core. Kim et al. [7] investigated the static strength of non-stitched and stitched polyurethane foam-cored sandwich structures. The experimental results indicated that the bending strength of the stitched specimen increased by 50% in contrast with the non-stitched specimen. The strength of stitched specimen increased with the stitching thread diameter and decreased with the stitching thread distance. Potluri et al. [8] developed a novel stitch-bonding technique using tufting and weft insertion techniques to accommodate the relatively hard and thick core materials. The interface strength of face sheet/core was improved and the total energy absorption and stiffness of the sandwich panels increased with the stitch density. Lascoup et al. [9] proposed a monolithic stitched strategy to strengthen the face sheet/core interface of a composite sandwich structure. The beneficial profits in bending, shearing, and compression performance were obtained. Although the mass increment of the stitched panels was unavoidable, the specific performance still remained considerable. At the same time, Lascoup et al. [10] evaluated the impact resistance of sandwich structures consisting of glass yarn stitched polyurethane foam core. The advantages of a stitched core sandwich, compared with the non-stitched core sandwich, were the maximal effort and rigidity. The delamination failure mode, which often occur in traditional non-stitched sandwich structures, was not observed. Han et al. [11] showed that the stitched sandwich configuration improved the impact resistance and energy absorption by inhibiting the crack expansion and therefore improving the strength of the core. Shigang et al. [12] also found that much improvement of the tensile and inter-laminar shear stiffness and slight improvement of bending stiffness were achieved by stitching the sandwich panel, whereas the tensile and inter-laminar shear stiffness were more sensitive to the stitching angle, and 45 degrees was recommended for the stitching angle. Hu et al. [13] experimentally studied the effect of different suture spacing of stitched fiberglass of the sandwich panel on the strain energy release of a mode I fracture. The results showed that the peak load and fracture displacement changed with suture spacing, and an optimal dimension of 8 mm was recommended for the suture spacing.

Regarding the core machining with holes and channels, the inner core of sandwich structures is machined in advance with various holes and channels; the core is then laid on the mold together with the face sheets; finally, the sandwich structures are infused and cured with liquid resin. As a result, the resin columns filling in the holes and channels are imported into the core layer. Compared with the stitching and z-pinning reinforcement techniques, the approach of core machining with holes and channels has three advantages: (i) the core can be easily machined with various geometries, therefore the machining cost of the core will be relatively low and the architecture of resin column can be versatile; (ii) the matching ability between sandwich structures and the corresponding mold can be improved because the flexibility of the core can be adjusted through thick channels; and (iii) the resin flows in the in-plane and through-thickness directions are easier to regulate through surface channels and through-thickness holes. Obviously, the presence of resin columns with high stiffness and strength would unavoidably affect the performance of the cores and sandwich structures [14,15]. In this field, few authors have published valuable references.

Truxel et al. [16] designed five resin infusion strategies and studied these treatments on debonding resistance of glass fiber/vinyl ester face sheets/balsa wood and PVC foam cores by the tilted sandwich debond experimental method. The results showed that grooved cores can enhance the local fracture resistance, because the transverse grooves can block the crack growing rapidly, and thick grooves exhibit remarkable resistance to crack propagation. On the contrary, the region without grooves exhibits lower face/core fracture toughness due to the lack of resin. The specimens with a layer of continuous filament mat between the face sheet and core show the highest fracture toughness among all the strategies.

Massüger and Gtzi [17] designed sandwich structures with fiberglass/polyester composite face sheets and contour cut, groove, and perforated hole (GP), and asymmetric double cut PET cores, and then examined the effect of the core machining configurations on the resin absorption and fatigue resistance. The results revealed that the resin absorption of GP core is about 70% and 50% for 20 mm and 30 mm thickness foam, respectively. The machining configurations of the groove and the perforated hole had no negative effect on the fatigue resistance of the infused core, compared with the plain core, and the residual static shear strength remained 60% for plain and GP cores after 10 million load cycles. Berger et al. [18] also came to a similar finding that through-the-thickness groove had no negative effect on the fatigue resistance if the grooves are fully filled by resin.

May-Pat et al. [19] investigated the perforated core configuration on the through-the-thickness compression, plate shear, face sheet/core fracture toughness, and resin absorption of glass/polyester face sheets and a low-density PVC foam core sandwich panel. The results indicated that the compressive modulus of the sandwich structures increased greatly and shear modulus increased moderately under the condition of a 28% increase in weight. Debonding fracture toughness was enhanced slightly under the condition of a small fraction of perforated holes.

Mitra and Raja [20] proposed an innovative treatment of semi-circular glass-fiber/epoxy resin shear keys embedded between the E-glass/epoxy face sheet and the low-density PVC foam core for the purposes of upgrading the face/core debonding resistance under in-plane compressive loading. The result showed that the delamination resistance capacity increased by approximately 25%.

Halimi et al. [21] designed six different perforated hole patterns to examine hole distribution on the manufacturing parameters and mechanical properties of sandwich structures composed of E-glass fiber/polyester resin face sheets and a rigid PVC foam core. The results showed that the resin filling time of composite sandwich preform can be reduced by up to 40%, the bending load and yield energy absorption increased by 38% and 100%, respectively, along with a low weight increment of 3.6%. Furthermore, the critical debonding load and fracture toughness were moderately improved.

Fathi et al. [22] experimentally studied the flexural properties of sandwich structures made of glass fiber-reinforced/epoxy face sheets and various core materials including balsa wood, PET, PVC, and PU foams. The effects of core machining configuration on the bending stiffness, shear strength, and energy absorption were investigated. The results revealed that the contour cut fully filled with resin enhanced the shear strength. Some foams, in contrast with balsa wood, have higher energy absorption before failure although the specific shear strength of them was comparable. In addition, polymer foams were less discrete in density/properties for resin uptake during the manufacturing process.

Yokozeki and Iwamoto [23] investigated the effects of core machining configuration on the face sheet/core debonding toughness of carbon fiber reinforced epoxy/Divinycell HT70 sandwich panels by double cantilever beam tests. The specimens with perforated holes and grooves had higher face sheet/core interfacial debonding toughness, specifically for specimens with a perforated core.

Yalkin et al. [24] experimentally investigated the influence of the core machining of the plain, perforated hole, and glass fiber stitch on the compressive, bending, shear, and impact properties of composite sandwich structures. The results revealed that the mechanical performance of sandwich composites with perforated core can be improved compared with that of the plain foam core, and the mechanical performance of sandwich composites with glass fiber stitched foam core can be improved more significantly with less weight increase. The columns filled with resin or glass fiber composite blocked the movement of the foam core, and thereupon decreased the bending deflection and shear strain at failure.

Yalkin et al. [25] designed eight core configurations to experimentally investigate the flatwise tensile and compressive strength of sandwich structures. The results showed that the sandwich structure with glass-stitched core had the highest compressive strength, whereas the sandwich structure with glass-stitched to the face sheet had the highest tensile strength among all the configurations.

Balıkoğlu et al. [26] investigated the effect of perforated and grooved foam core modification on the bending performance of composite sandwich beams made of E-glass fibre/vinylester resin composite face sheets and closed cell PVC foam. The experimental results showed that the bending strength and effective stiffness of sandwich beam increased by 34% and 61%, respectively, compared with that without core modification. The perforation modification exhibited more effective enhancement of the extreme bending strength of sandwich beam than the grooved core.

From the previous studies, it can be summarized that: (i) some core machining configurations have been adopted such as perforate hole, shallow groove, contour cut, contour cut and stitch, but the combinations of core machining configurations were rarely adopted in previous tests; (ii) most studies only estimate one or two mechanical properties but lack a comprehensive knowledge of the effects on the various mechanical properties; (iii) quantitative evaluation on the mechanical properties of composite sandwich structures was rarely performed in consideration of the resin absorption and cost. To expand the understanding of the effects of core machining configurations on the mechanical properties and cost of the infused cores, three types of core machining configurations were adopted, and the tensile, compressive, shear, and peel tests were performed according to general standards. The modulus, strength, and corresponding strain, failure modes, and the cost of the infused core were analyzed, the effects of the core machining configurations on the mechanical properties were quantitatively estimated considering the resin absorption and material cost. The findings of this study would provide a comprehensive and quantitative insight into the effects of core machining configurations on the mechanical properties of the infused core and sandwich structure subjected to various loads. This could be helpful for further core machining design.

In the following section, the main materials, core machining configurations, design and manufacturing process for specimens, and test setups and apparatuses are elaborated. In Section 3, the measured compressive, tensile, and shear modulus and strength are comparatively discussed with the analytical method, and the compressive, tensile, shear, and peel strength, strain, and corresponding failure characteristics are analyzed. The cost merits of the cores after infused resin are also evaluated. Finally, conclusions are drawn in Section 4.

## 2. Experiment

### 2.1. Materials

The HPE110 foam from NMG COMPOSITES Co., Ltd. (Jiaxing, China) was adopted in consideration of the properties and cost, and is composed of a stiff polyurethane and a flexible polyether using interpenetrating network structure. The density of the foam is approximately 0.117 g/cm^3^. The epoxy resin (DQ200E/DQ204H) was from Sichuan Dongshu New Material Co., Ltd. (Deyang, China), which is one of the leading suppliers of epoxy resin. The ratio of the host agent DQ200E to the hardener DQ204H was specified to be 100:32 in weight, and the density of the cured epoxy resin is 1.157 g/cm^3^. The mechanical properties of the epoxy resin and foam are shown in Figure 1. The data for the epoxy resin and HPE110 foam were supplied by the manufacturers. The tri-axial glass fiber ETLX 1175 from Changzhou Tianchang Fiberglass Composites Co., Ltd. (Changzhou, China), was used to fabricate the face sheets of the sandwich structure, which consist of 567 g/m^2^ fiber in 0° direction, and 301 g/m^2^ fiber in 45° and −45° direction, respectively. The tensile strength, modulus, density, and the corresponding fiber content in mass of the fiberglass/epoxy resin composite material are 589.57 MPa, 28,930 MPa, 1.96 g/cm^3^, and 73%, respectively. The adhesive (WD3705) for bonding the specimens to the fixture is an epoxy paste from Shanghai KangDa New Materials Co., Ltd. (Shanghai, China).

### 2.2. Specimens and Manufacturing

#### 2.2.1. Core Machining Configurations

Three types of machining configurations were adopted for the cores, including: (i) plain core, (ii) grooved and perforated hole (GP) core, and (iii) grooved, perforated hole, and contour cut (GPC) core, respectively. The dimensions of the groove, perforated hole, contour cut, and their spacings are shown in Figure 2 and Table 1. It should be noted that the contour cut width was determined by the resin absorption test.

#### 2.2.2. Specimen Design

The tensile, compressive, shear, and peel tests were carried out according to ASTM C297-2016 [27], ASTM C365-2016 [28], ASTM C273-2016 [29], and ASTM D1781-98(2012) [30], respectively. The dimensions of the test specimens are shown in Table 2. Five specimens were prepared for each tensile, compressive, and shear test, and three specimens for the peel test.

#### 2.2.3. Specimen Manufacturing

The vacuum-assistant infusion process was adopted to fabricate the infused cores and sandwich plates. The layup layout of the main materials used in the sandwich structures and the required accessories are shown in Figure 3, which would be infused with epoxy resin. After 6 h of infusion and curing at 45 °C and 8 h of post-curing at 70 °C, the expected core layer infused resin and sandwich structure was obtained by removing the outside layer of the release film. Finally, the plates were cut according to the required dimensions of test specimens.

For the compressive and peel test specimens, a bonding procedure between the specimen and additional fixtures was not needed because the test specimen can be directly fixed between the clamps under compressive and peel loadings. For the tensile and shear test specimens, two extra fixtures were needed to be bonded on the top and bottom of the specimen for the convenience of applying loads. The bonding process was carried out in four steps as follows: (i) sanding and clearing the surface of the fixture to form a clean and rough bonding surface; (ii) mixing the adhesive paste and putting it on the surfaces of the specimen and fixture, then clamping the fixtures and specimen together with the help of an aligning device; (iii) moving the fixtures and specimen together into the cabinet; and (iv) maintaining a constant temperature of 40 °C for 24 h.

Four big GP and GPC core plates were used to obtain the resin absorption by measuring the weight change after the resin infusion. The average resin absorption weights were 1181.2 g/m^2^ and 2564.9 g/m^2^ for typical 25-mm thick GP and GPC cores, respectively. Assuming identical resin layer thickness for all core surfaces, the equivalent thickness of the resin layer after the foam infused resin can be calculated by the resin absorption of the GP specimens, as shown in Table 1; the width of the contour cut therefore can be calculated through the resin absorption of the GPC specimens. It can be seen that the equivalent thickness of resin layer filled in the surface cell of the foam was 0.17 mm, which is not too small to be neglected. Therefore, the equivalent thickness of the resin layer would be considered in the following analytical prediction for the strength and moduli of infused cores and cost evaluation of the resin.

### 2.3. Test Setups and Apparatuses

The universal material testing machine (MTS CMT 5105, 100 kN) was used in all mechanical tests. The tensile and compressive strains (YYU-5/25 and YY25/3 extensometers), shear strain (12.5 mm micrometer), and peel displacements (crosshead displacement) were recorded during the tests, as shown in Figure 4. The corresponding extensometer and micrometer were from Central Iron & Steel Research Institute, Changchun Sanjing Testing Instrument Co., Ltd. (Beijing, China), and SWISS-SYLVAC (Yverdon-les-Bains, Switzerland), respectively. The rubber ropes were used to fix the extensometer at the middle region during the tensile and compressive tests, and the micrometer was used to measure the relative displacement of the two fixtures during the shear test.

The loading speeds were 1.0, 2.0, 1.0, and 25.4 mm/min for tensile, compressive, shear, and peel tests, respectively. The ambient temperature and relative humidity during the test period were 20–22 °C and 47–52%, respectively.

## 3. Results and Analysis

### 3.1. Compressive Test

Figure 5a shows the compressive moduli of the plain, GP, and GPC cores. It can be seen that the compressive moduli of the GP and GPC cores increased by 66.69% and 214.08%, respectively, compared with the plain core. Infused resin has great impact on the compressive moduli of the cores because it has far higher modulus than the foam.

A parallel model [31] was adopted to predict the compressive moduli of infused cores, and the calculating procedures are shown in Appendix A. The predicted compressive moduli of the GP and GPC cores were 95.95 MPa and 254.55 MPa, respectively. The deviations of the compressive moduli were −7.9% and 29.7% for the GP and GPC cores, respectively. It can be concluded that the analytical model can well estimate the compressive modulus of GP core infused resin. For the GPC core, the analytical model seems to overestimate the compressive modulus. May-Pat et al. [19] also noted a similar trend that the compressive modulus of the GP core increased significantly compared with the plain core, but the parallel model adopted could not predict the compressive modulus accurately for low-density PVC core.

Figure 5b,c show the compressive strength and strain at peak load of the plain, GP, and GPC cores. The compressive strength of the GP and GPC cores increased by 51.8% and 301.4% compared with the plain core, respectively. The corresponding predicted strength by the analytical model was 2.36 MPa and 6.97 MPa, respectively, as shown in Appendix A. It can be concluded that the infused resin can improve the compressive strength of the GP and GPC cores substantially, especially for the GPC core, and the strength of the infused core can be approximately predicted according to the unified strain in the foam and infused resin.

Moreover, the consistency of the experimental and predicted compressive strength was validated, indicating that the compressive strains at peak load for the plain, GP, and GPC cores were very close. Therefore, the strength of the foam and the infused resin can be used to a large extent, as shown in Figure 5b.

Figure 6a shows the corresponding fracture morphology of the GP core. The fracture morphology of failure specimens was not easy to observe from the outer surface of the specimen because the failure morphology of the pure foam specimen under compressive failure load could not be observed by the naked eye. The compressive crush of infused resin columns with a brittle rupture sound was captured continuously, but it left no fracture surface on the outside of the specimens. Figure 6b shows the corresponding fracture morphology of the GPC core. The stiff resin columns in contour cut penetrated the foam and made the foam/resin column debonding, the resin column buckled and broke around the tip without the support of the foam, and finally the foam could not sustain the loads alone and started to crush after the failure of the resin columns.

### 3.2. Tensile Test

Figure 7a shows the tensile moduli of the plain, GP, and GPC cores. The tensile moduli of the GP and GPC cores increased by 48.2% and 185.3%, respectively, compared with the plain core. The parallel model was also adopted to predict the tensile moduli of infused cores (see Appendix A). The calculated tensile moduli of the GP and GPC cores were 102.18 MPa and 260.45 MPa, respectively. The analytical model can accurately predict the tensile modulus of the GP core but overestimated the tensile modulus of the GPC core.

Figure 7b,c show the tensile strength and strain at peak load of the plain, GP, and GPC cores. Figure 8 shows the corresponding fracture morphology of the GP and GPC cores. It can be seen that the tensile strength of the GP and GPC cores changes slightly compared with the plain core; in other words, the import of the infused resin made no contribution to the tensile strength of the GP and GPC cores. The predicted tensile strength of the GP and GPC cores was 2.12 MPa and 5.27 MPa, respectively, which were much greater than the corresponding experimental values. The analytical model overestimated the tensile strengths for both GP and GPC cores, especially for the GPC core.

Except for one GPC specimen debonding at the adhesive/fixture interface (not included in the analysis), all GP and GPC specimens failed within the infused core (shown in Figure 8). The resin column initially broke near the bonding region, which led to the crack propagation across the foam nearby, and further resulted in the breakage of the foam and resin column far away. Moreover, the strain of the GP and GPC cores at peak load were far less than that of the epoxy resin or foam separately, that the premature failure occurred at small strains. This phenomenon has two possible explanations: one is that an abrupt cross section of the resin column shaped by the irregular foam cell tends to bring about stress concentration, and worse, the brittle materials, such as epoxy resin, are more susceptible to stress concentration under tensile load than compressive load [32,33], which resulted in the premature failure of resin column far below the tensile strength from the standard specimen. This explanation could be proved by the measured strain at peak load, which decreased sharply after infusion resin, as shown in Figure 7c. The other explanation is the shear-lag phenomenon of the foam/resin column interface [34], which was observed in Figure 8 when the resin columns were pulled out.

### 3.3. Shear Test

Figure 9a shows the shear moduli of the plain, GP, and GPC cores. It can be seen that the shear moduli of the GP and GPC cores increased by 28.1% and 134.2% compared with the plain core, respectively, which indicates that the shear moduli of the cores can be improved by grooves, perforated holes, and contour cut machining. A similar finding was also obtained in the experiments by May-Pat et al. [19].

To calculate the shear moduli of the GP and GPC cores, the semi-experimental Halpin-Tai theory [35] and Nickel’s formula [36] were adopted. The calculated shear moduli were 28.85 MPa and 66.38 MPa for the GP and GPC cores, respectively, as shown in Appendix A, and the deviation of the shear moduli for the GP and GPC cores were −0.3% and 25.4%, respectively. It can be seen that the analytical model can well predict the shear modulus of the GP core but overestimated the shear modulus of the GPC core.

Figure 9b shows the shear strength of the plain, GP, and GPC cores. It can be seen that the shear strength increased moderately after infused resin. May-Pat et al. [19] also found that the shear strength of the GP core did not change obviously compared with the plain core. It should be noted that the GPC core had a little bit lower shear strength even though it was infused with more resin than that of the GP core.

Figure 10 shows the fracture morphology of the shear test specimens. It can be seen that all the GP and GPC specimens failed near the interface between the foam and resin. It is worth noting that the failure surfaces were at the non-groove sides for all GP specimens, but at groove sides for all GPC specimens. That is to say, the shear strength at the groove side surfaces for GP specimens was stronger than that without grooves, whereas the shear strength at the non-groove side surfaces for GPC specimens was stronger than that without grooves. The surface between contour cut tips and grooves for the GPC became a weak region, so that the shear strength of the GP core was even slightly greater than that of the GPC core. Fathi et al. [22] found the asymmetric double cut has a negative effect on the shear strength of PET sandwich structures in a bending test, but a positive effect for PVC sandwich structures. Based on previous studies, it can be concluded that the shear strength depended much on the surface morphology of the foam core, and the through-thickness cut or perforated hole can help to improve the shear strength.

### 3.4. Peel Test

Figure 11a shows the composite face sheet/core peel strength of the plain, GP, and GPC core sandwich structures. It can be seen that the peel strength of sandwich structures with GP and GPC cores increased remarkably compared with that with the plain core. The peel strength of the interfaces of infused cores with perforated holes and grooves (of GP), sparser perforated holes (of GP), perforated holes and grooves (of GPC), and perforated holes and contour cut (of GPC), increased by 53.16%, 38.75%, 51.43%, and 135.69%, respectively. It is clear that the specimens with contour cut surface have the highest peel resistance among all the core machining configurations, followed by the specimens with perforated holes, and the specimens with grooves have the lowest peel resistance. With regard to the bonding area of the core machining to the face sheet, the grooves lead to the largest area (19.00% for GP and GPC cores), followed by the contour cut (5.72%), and the perforated hole is the smallest (1.08% and 0.48% for GP and GPC cores, respectively). The least-square method was utilized to obtain the peel strength of the individual core machining configurations with minimal error under the condition of over-determined equations. The calculated results are shown in Figure 11b, and it can be seen that the specimen with perforated hole had the highest efficiency of peel resistance (590.29 N.mm/mm), followed by the specimen with contour cut (439.06 N.mm/mm), and the specimen with the shallow groove had the worst performance (49.03 N.mm/mm). The efficiency of the shallow grooves was far behind the perforated hole and contour cut. In previous studies, similar work had been done by a few authors [16,20,21,23], they also found the positive effects of these core machining configurations on peel strength. However, they did not quantitatively and comprehensively evaluate the peel strength of sandwich structures so were unable to get a full understanding on the effects of the core machining configurations.

Figure 12a exhibits the photography of fracture surface of peel specimens. The pits of fracture surface belonging to the face sheet emerged at the locations of resin grooves and gaps and at the raised foam around the pits, and the opposite morphology emerged at the fracture surface belonging to the core. To clarify the micro-topography of the fracture surface, an optical microscope was used to capture the fracture photography magnified by 212 times; the fracture photography of perforated hole, cut gap infused resin, and foam are shown in Figure 12b–d. The fracture without resin column exhibited cohesive failure of the core foam near the interface between the core foam and the face sheet. The indentations in Figure 12d,e indicate that the fracture occurred at the interface between the resin column of the cores and fibers of the face sheet [37]. It can be explained that the debonding strength of epoxy resin and fibers (36.6 MPa) [38] is lower than the tensile strength of the pure resin (68.39 MPa) but much greater than the tensile strength of the pure foam (1.49 MPa).

### 3.5. Cost Evaluation

The price of the foam and epoxy resin are 852.00 $/m^3^ and 4662.71 $/m^3^, respectively. A typical thickness of 25 mm, which was commonly adopted in [15,19,22,23,26], was decided as the target thickness of the foam for evaluating the cost of infused cores. The fractions of infused resin were 0.00%, 1.08%, and 6.20% in volume for the plain, GP, and GPC cores, respectively. The relative merit $m_{r}$ is defined by Equation (1) to evaluate the mechanical properties in consideration of the material cost. The results were shown in Figure 13.
(1)$$m_{r}=\frac{P_{i\_c}\times C_{p\_c}}{C_{i\_c}\times P_{p\_c}}.$$

It can be seen that all the GP and GPC cores exhibit a positive effect in compressive, tensile, and shear moduli as well as compressive strength, except that the GPC core exhibits a negative effect on tensile strength. The GPC machining configuration had a larger positive or negative effect on the above properties of the core than that of the GP machining configuration because the resin fraction of the GPC core was larger than that of the GP core in volume. None of the previous studies conducted similar evaluations. The evaluation on the mechanical properties of infused cores in consideration of the cost of the resin used provides a practical and quantitative method to estimate the return in mechanical properties versus material cost input.

## 4. Conclusions

Core machining configurations are commonly used to improve the bendability and facilitate the resin infusion during the manufacturing process of composite sandwich structures. The resin uptake in the holes and channels of the cores after machining can change the mechanical properties of the cores and corresponding sandwich structures. This study aims at making clear the effect of core machining on the mechanical properties and cost of the cores and corresponding sandwich structures. Static tensile, compressive, shear, and peel tests were performed on the raw foam, infused cores, and the corresponding sandwich structures. The findings are as follows: (1)The GP and GPC machining configurations can greatly improve the tensile, compressive, and shear moduli, and compressive strength of the cores in the condition of a small volume fraction of infused resin. Analytical models can predict these mechanical properties with desired accuracy, but fail to estimate the tensile strength due to stress concentration.(2)The tensile strength of the infused GP and GPC cores was barely improved compared with the pure foam core. This was due to the stress concentration of abrupt resin cross-section shaped by the irregular foam cell and the shear-lag phenomenon of the resin column/foam interface.(3)The shear strength of the infused GP and GPC cores increased in a small amount compared with the pure foam core, which was determined by local structure around the face sheet/core interface. The comparative strength of the resin, the foam, and the fiber/matrix interface determined the fracture surface and the final shear strength.(4)Thick resin columns filled in the perforated holes and contour cuts had relatively large face sheet/core peel capacity, and thin resin channels filled in the grooves had limited improvement on the peel capacity.(5)Both GP and GPC cores gained positive effects on the compressive, tensile, and shear moduli, and compressive strength in consideration of the core cost after resin infusion, but the GPC core exhibited a negative effect in tensile strength.

In the future, the optimization design of the dimensions and progressive failure analysis will be conducted in terms of the core machining configurations and corresponding sandwich structures for the purposes of in-depth understanding the effects of core modifications on the performance of composite sandwich structures.

## Figures and Tables

**Figure 1 materials-15-00521-f001:**
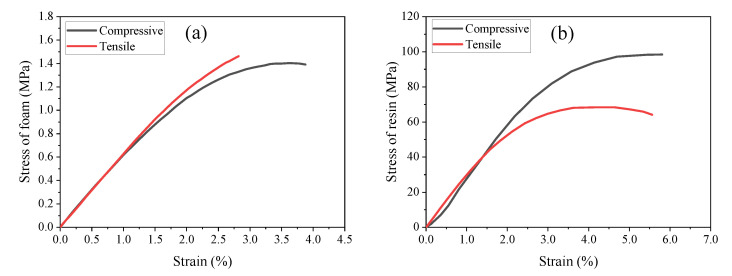
The stress-strain curve of the materials: (**a**) HPE110 foam and (**b**) DQ200E/DQ204H epoxy resin.

**Figure 2 materials-15-00521-f002:**
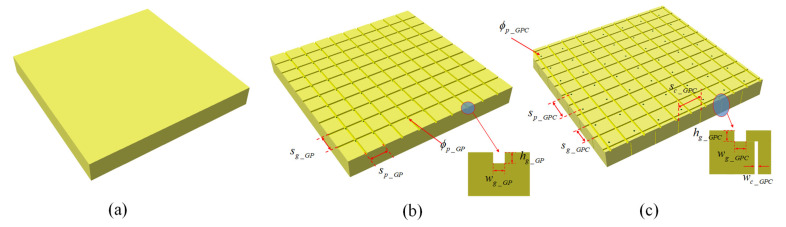
Core machining configurations: (**a**) plain core, (**b**) GP core, and (**c**) GPC core.

**Figure 3 materials-15-00521-f003:**
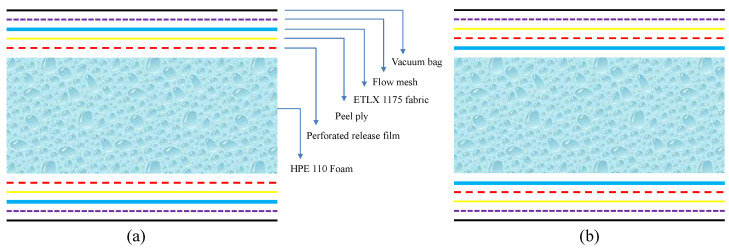
The layup layout for the specimens: (**a**) infused core and (**b**) sandwich structure.

**Figure 4 materials-15-00521-f004:**
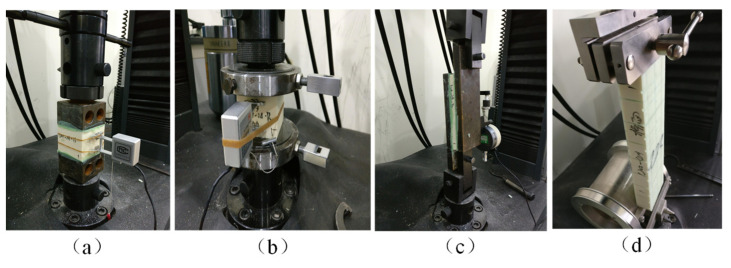
The in-site pictures of the tests: (**a**) tension, (**b**) compression, (**c**) shear, and (**d**) drum peel.

**Figure 5 materials-15-00521-f005:**
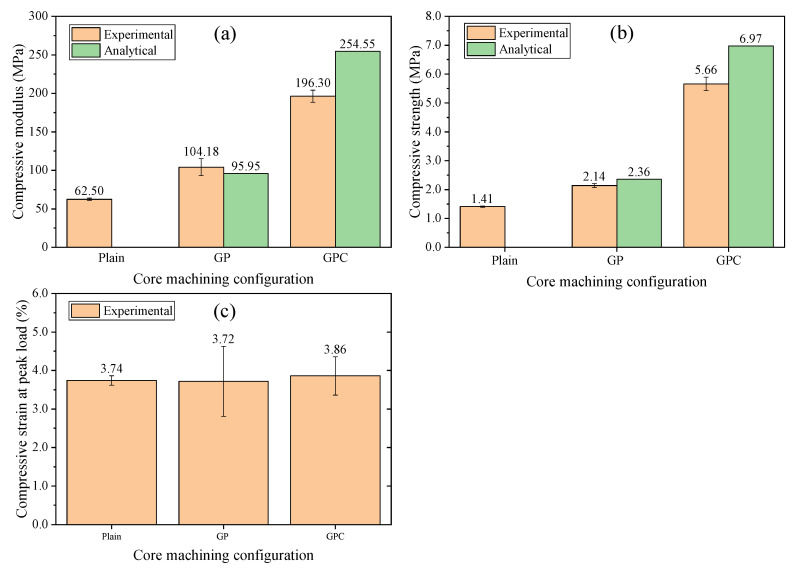
The compressive properties of the plain, GP, and GPC cores: (**a**) modulus, (**b**) strength, and (**c**) strain at peak load.

**Figure 6 materials-15-00521-f006:**
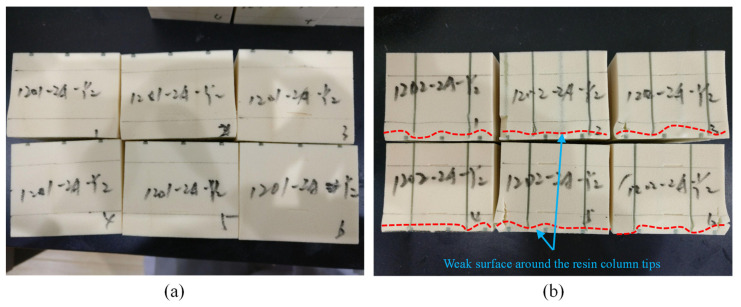
Failure modes of the compressive specimens: (**a**) GP core and (**b**) GPC core.

**Figure 7 materials-15-00521-f007:**
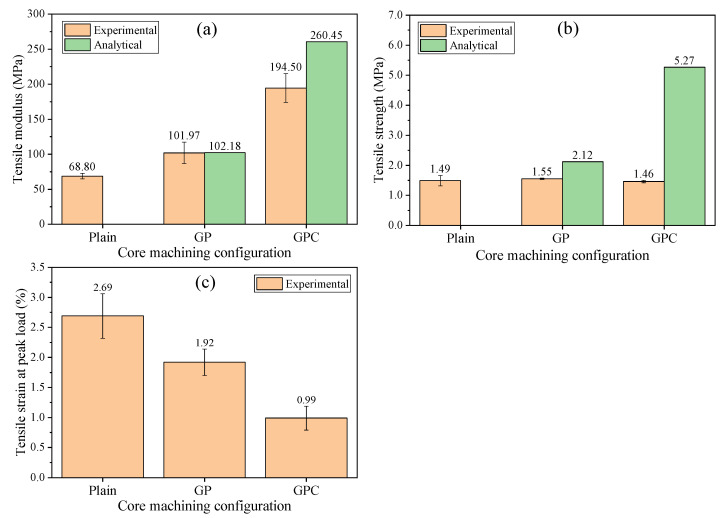
The tensile properties of the plain, GP, and GPC cores: (**a**) modulus, (**b**) strength, and (**c**) strain at the peak load.

**Figure 8 materials-15-00521-f008:**
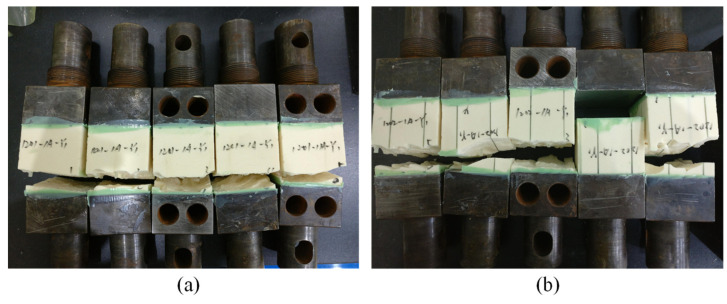
Failure modes of the tensile specimens: (**a**) GP core and (**b**) GPC core.

**Figure 9 materials-15-00521-f009:**
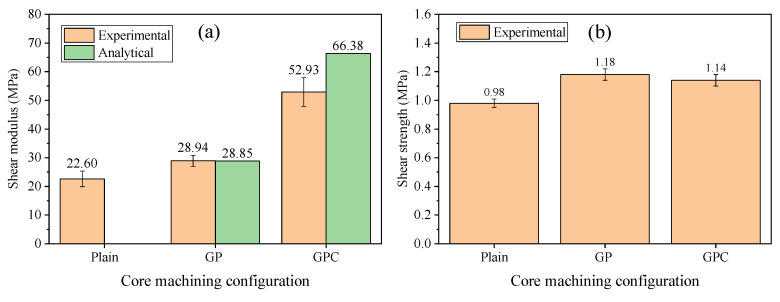
The shear properties of the plain, GP, and GPC cores: (**a**) modulus and (**b**) strength.

**Figure 10 materials-15-00521-f010:**
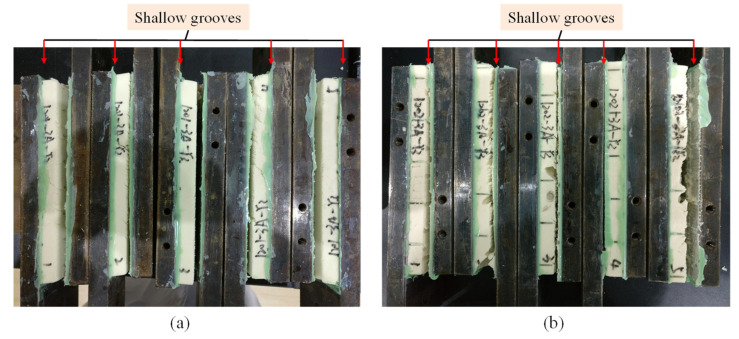
Failure modes of the shear specimens: (**a**) GP core and (**b**) GPC core.

**Figure 11 materials-15-00521-f011:**
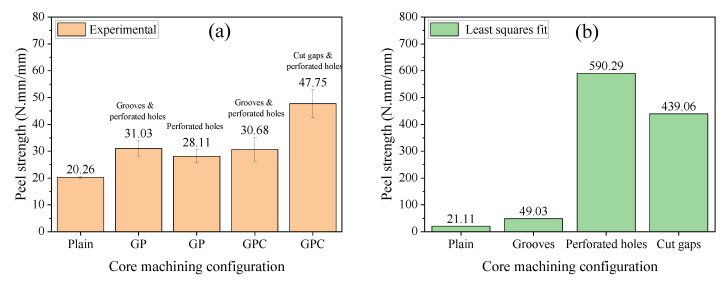
The peel strength of the cores: (**a**) the plain, GP, and GPC machining configurations and (**b**) the individual core machining configuration.

**Figure 12 materials-15-00521-f012:**
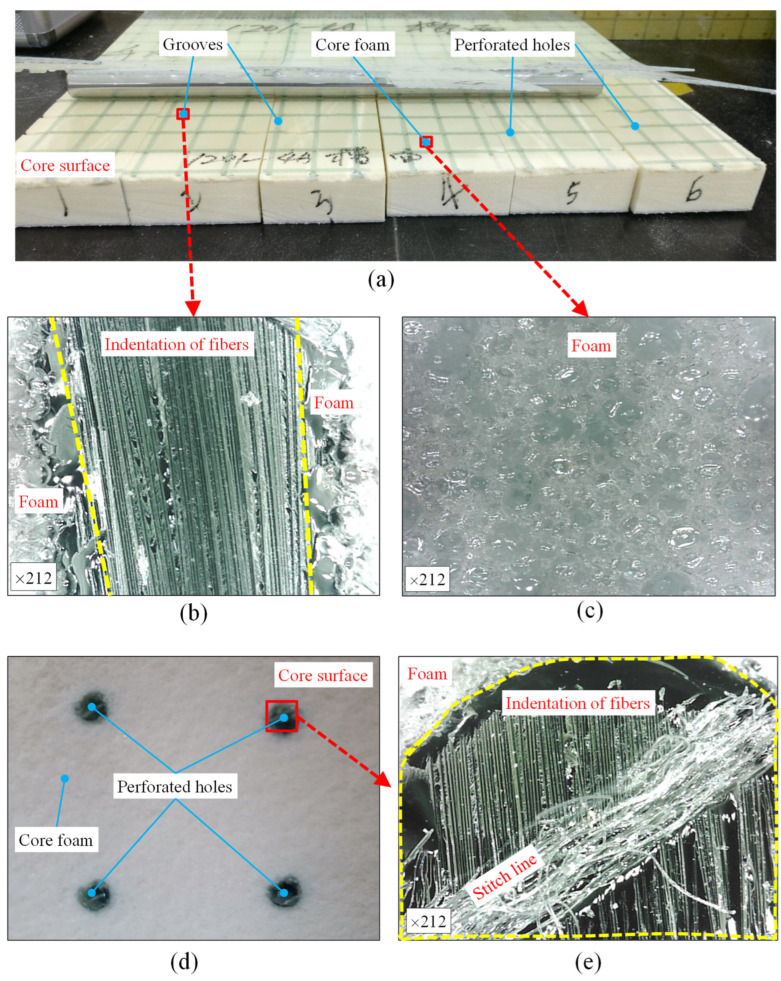
Typical fracture morphology of the peel specimens: (**a**) fracture surfaces of GP core specimens, (**b**) the micro-topography of grooves, (**c**) the micro-topography of foam, (**d**) the other fracture surface of the GP core specimen, and (**e**) the micro-topography of perforated hole.

**Figure 13 materials-15-00521-f013:**
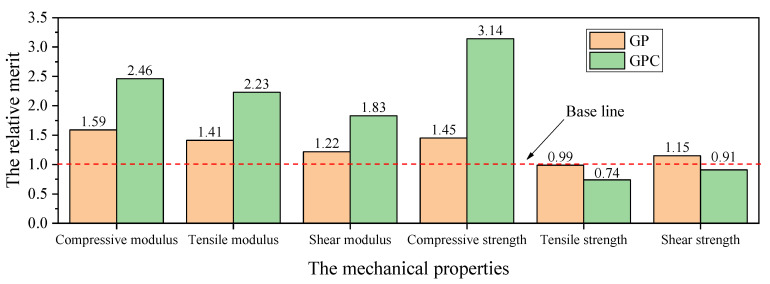
The relative merit corresponding to mechanical properties.

**Table 1 materials-15-00521-t001:** The dimensions of the groove, perforated hole, contour cut, and their spacings.

**Nomenclature**	$s_{g\_GP}$	$h_{g\_GP}$	$w_{g\_GP}$	$s_{p\_GP}$	$\phi_{p\_GP}$	$t_{a\_c}$	
Value/mm	20 × 20	2.0	2.0	20 × 20	2.0	0.170	
**Nomenclature**	$s_{g\_GPC}$	$h_{g\_GPC}$	$w_{g\_GPC}$	$s_{p\_GPC}$	$\phi_{p\_GPC}$	$s_{c\_GPC}$	$w_{c\_GPC}$
Value/mm	20 × 20	2.0	2.0	30 × 30	2.0	30 × 30	0.531

**Table 2 materials-15-00521-t002:** The dimensions of the test specimens.

Test Types	Dimensions (Length × Width × Height)
Tensile test	60 mm × 60 mm × 50 mm
Compressive test	60 mm × 60 mm × 50 mm
Shear test	150 mm × 60 mm × 12.5 mm
Peel test	240 + (2 × 30) mm × 60 mm × 25 mm

## Data Availability

Data are available on request at corresponding authors.

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
