# Peer review of "The Effects of Core Machining Configurations on the Mechanical Properties of Cores and Sandwich Structures"

_materials, 2022, doi:10.3390/ma15020521_

Round 1

Reviewer 1 Report

This paper presents a series of results obtained in testing the properties of sandwich composites

The authors can consider the following aspects:

- The introduction needs to be substantially improved in the sense that other bibliographic sources in the field need to be considered. The first two paragraphs of the paper are not directly related to the research presented in the paper;

- Figure 1 from introduction should be deleted;

- Research methodology and research objective are unclear. A series of size values ​​were established for 3 types of cores. What are the justifications behind those dimensions? Several values ​​should have been chosen for the dimensions presented in order to achieve an optimization of the geometric parameters of the cores;

- In order to be able to achieve an optimization of the geometric parameters of the cores it is necessary to perform a detailed statistical processing of the results obtained in the experimental research;

- a more detailed microscopic analysis of the test tube material is required both before and after testing;

- the graphical processing of the experimental results is a rudimentary one

- The discussion part needs to be further developed in order to highlight the contribution of the research presented in the paper in relation to other research in the field;

- At the end of the conclusions, the future research directions should be specified;

- the conclusions should be more concrete and cover the practical applications of the results obtained.

Author Response

Dear reviewers,

Thank you for your comments and suggestions in regard with the manuscript entitled " The effects of core machining configurations on the mechanical properties of cores and sandwich structures".

We are appreciated for your selfless efforts on reviewing the manuscript carefully. All comments and suggestions are valuable and helpful to improve the manuscript. In full consideration of the comments and suggestions, we have made corresponding corrections (marked red) and detailed explanations one by one. We hope the revised manuscript and the explanations meet your expectation.

Please find the attachments:

(Ñ–) Response letter,

(Ñ–i) Revised manuscript (with changes marked),

Reviewer 2 Report

This paper investigates e the comprehensive mechanical properties of the cores and sandwich structures through core modification.
The paper is well structured, Figures and tables are clear.
Some minor comments and suggestions are listed as follow:
The introduction should be enriched with State of the art paper where authors investigate the mechanical properties of different kinds of foam:
10.1016/j.tafmec.2021.102924
10.1016/j.tafmec.2019.03.001
10.1007/s11340-011-9519-7
I recommend introducing an outline of the paper at the end of the introduction section.
A discussion about a possible anisotropy of the specimens is recommended.
I recommend t mention the standard adopted for each test.
Conclusion sections appear clear.
The journal template has been not adopted.

Author Response

Dear  reviewers,

Thank you for your comments and suggestions in regard with the manuscript entitled " The effects of core machining configurations on the mechanical properties of cores and sandwich structures".

We are appreciated for your selfless efforts on reviewing the manuscript carefully. All comments and suggestions are valuable and helpful to improve the manuscript. In full consideration of the comments and suggestions, we have made corresponding corrections (marked red) and detailed explanations one by one. We hope the revised manuscript and the explanations meet your expectation.

Please find the attachments:

(Ñ–) Revised manuscript (with changes marked),

Comment #2-1

English language and style: (x) Moderate English changes required

Response to comment #2-1

Thank you very much for your positive feedback and constructive suggestion.

We have made a thorough revision of the paper, including language, format, phrases, figures, and structural arrangement, as shown in the red parts of the revised manuscript.

Comment #2-2

This paper investigates the comprehensive mechanical properties of the cores and sandwich structures through core modification.
The paper is well structured, Figures and tables are clear.

Response to comment #2-2

Thank you very much for your positive feedback and valuable suggestion on the manuscript. Your encouragement will become our motivation for further work.

Comment #2-3

Some minor comments and suggestions are listed as follow:
The introduction should be enriched with State of the art paper where authors investigate the mechanical properties of different kinds of foam:
10.1016/j.tafmec.2021.102924
10.1016/j.tafmec.2019.03.001
10.1007/s11340-011-9519-7

Response to comment #2-3

Thank you very much for your strict review and constructive suggestions.

We have re-written the beginning of the introduction, expanded related bibliographic sources with the guide of your suggestions. Please find the revised manuscript.

In addition, we have reviewed the research progress of the other two strategies (Z-pinning and stitching) for reinforcing the cores and sandwich structures because journal articles related to the core machining configuration are limited.

Comment #2-4

I recommend introducing an outline of the paper at the end of the introduction section.

Response to comment #2-4

Thank you for your constructive suggestion.

The structural arrangement of the paper has been added at the end of the Introduction, please find the revised manuscript. The added content is also shown in the following paragraph.

In the following section, the main materials, core machining configurations, design and manufacturing process for specimens, and test setups and apparatuses are elaborated. In Section 3, the measured compressive, tensile, and shear modulus and strength are comparatively discussed with the analytical method, the compressive, tensile, shear, and peel strength, strain, and corresponding failure characteristics are analyzed. Meanwhile, the cost merits of the cores after infused resin are evaluated. Finally, conclusions are drawn in Section 4.

Comment #2-5

A discussion about a possible anisotropy of the specimens is recommended.

Response to comment #2-5

Thank you for your constructive suggestion.

With regard to the possible anisotropy of the specimens, similar issues have not been discussed in the previous studies.

 In our opinion, the anisotropy of infused cores or the corresponding sandwich structures would be a significant issue. For a pure foam core, it is generally regarded as an isotropic material though the mechanical properties of the core have a slight difference between in in-plane and through-thickness directions. For an infused core with periodical distributed perforated holes or through-thickness cuts, it can be considered as an orthotropic material. For an infused core with combined perforated holes, grooves and contour cuts, the architecture of this material has no symmetry axis or symmetry planes, it fulfills the requirements of an anisotropic material. For the corresponding sandwich structures with the anisotropic cores, the architecture and mechanical property are too complex to make it clear in few sentences and worthy of further in-depth research separately.

In the current work, we focus on a comprehensive estimation of the GP and GPC core machining configurations on the mechanical properties of the cores and corresponding sandwich structures. In order to emphasize the above contents, the discussion on anisotropy of the specimens should be performed in a separate study for the purposes of a clear and full understanding of its effects.

Comment #2-6

I recommend t mention the standard adopted for each test.

Response to comment #2-6

Thank you for your constructive suggestion.

The international standards of tensile, compressive, shear, and peel tests for cores and sandwich structures have been added in the revised manuscript.

The corresponding contents in the revised manuscript are also shown in the following paragraph:

2.2.2. Specimen design

The tensile, compressive, shear, and peel tests were carried out according to the ASTM C297-2016 [27], ASTM C365-2016 [28], ASTM C273-2016 [29], and ASTM D1781-98(2012) [30], respectively. The dimensions of the test specimens are shown in Table 2. Five specimens were prepared for each tensile, compressive, and shear test, and three specimens for peel test.

Comment #2-7

Conclusion sections appear clear.
The journal template has been not adopted.

Response to comment #2-7

Thank you for your positive feedback and constructive suggestion.

The journal template has been adopted in the revised manuscript, please find the revised manuscript.

Round 2

Reviewer 1 Report

The authors revised their manuscript according to my suggestions. Thus the manuscript can be accepted for publication.

Reviewer 2 Report

the authors do not integrated the suggested literature.